# Epstein–Barr Virus Promotes Oral Squamous Cell Carcinoma Stemness through the Warburg Effect

**DOI:** 10.3390/ijms241814072

**Published:** 2023-09-14

**Authors:** Chukkris Heawchaiyaphum, Hironori Yoshiyama, Hisashi Iizasa, Ati Burassakarn, Zolzaya Tumurgan, Tipaya Ekalaksananan, Chamsai Pientong

**Affiliations:** 1Department of Biotechnology, Faculty of Science and Technology, Thammasat University (Rangsit Center), Pathum Thani 12120, Thailand; chukheaw@tu.ac.th; 2Department of Microbiology, Faculty of Medicine, Shimane University, Shimane 693-8501, Japan; iizasah@med.shimane-u.ac.jp (H.I.); atibura@kku.ac.th (A.B.); zolzaya.tumurgan@med.shimane-u.ac.jp (Z.T.); 3Department of Microbiology, Faculty of Medicine, Khon Kaen University, Khon Kaen 40002, Thailand; tipeka@kku.ac.th; 4HPV & EBV and Carcinogenesis Research Group, Khon Kaen University, Khon Kaen 40002, Thailand

**Keywords:** Epstein–Barr virus, oral squamous cell carcinoma, Warburg effect, glycolysis, cancer stem cell, CD44, GLUT1, LDHA

## Abstract

Epstein–Barr virus (EBV) is associated with various human malignancies. An association between EBV infection and oral squamous cell carcinoma (OSCC) has recently been reported. We established EBV-positive OSCC cells and demonstrated that EBV infection promoted OSCC progression. However, the mechanisms by which EBV promotes OSCC progression remain poorly understood. Therefore, we performed metabolic analyses of EBV-positive OSCC cells and established a xenograft model to investigate the viral contribution to OSCC progression. Here, we demonstrated that EBV infection induced mitochondrial stress by reducing the number of mitochondrial DNA (mtDNA) copies. Microarray data from EBV-positive OSCC cells showed altered expression of glycolysis-related genes, particularly the upregulation of key genes involved in the Warburg effect, including *LDHA*, *GLUT1*, and *PDK1*. Furthermore, lactate production and LDH activity were elevated in EBV-positive OSCC cells. EBV infection significantly upregulated the expression levels of cancer stem cell (CSC) markers such as *CD44* and *CD133* in the xenograft model. In this model, tumor growth was significantly increased in EBV-positive SCC25 cells compared with that in uninfected cells. Furthermore, tumorigenicity increased after serial passages of EBV-positive SCC25 tumors. This study revealed the oncogenic role of EBV in OSCC progression by inducing the Warburg effect and cancer stemness.

## 1. Introduction

Epstein–Barr virus (EBV) is a ubiquitous human gamma herpesvirus that causes persistent infections in more than 90% of the world’s population [1]. EBV is an oncogenic human herpes virus that accounts for 1.8% of all cancer-related deaths worldwide [2]. EBV infection is associated with various types of lymphocytic malignancies, including Burkitt’s lymphoma, Hodgkin’s lymphoma, and NK/T-cell lymphoma. EBV is also associated with epithelial malignancies, such as nasopharyngeal carcinoma (NPC) and EBV-associated gastric cancer (EBVaGC) [1,3,4].

Additionally, EBV is associated with other types of epithelial tumors such as lung cancer and oral squamous cell carcinoma (OSCC) [5,6]. She et al., reported that EBV infection increased the risk of OSCC by 5.03-fold [7]. We previously showed that EBV infection significantly promotes OSCC progression by inducing proliferation, migration, invasion, and suppression of apoptosis of the cells [6]. However, the underlying molecular mechanisms by which EBV drives OSCC carcinogenesis and tumorigenesis remain largely unknown.

Metabolic reprogramming is a critical hallmark of cancer development. Alterations in metabolic programming can modulate malignant phenotypes and the tumor microenvironment [8]. Aerobic glycolysis, also known as the Warburg effect, is common in cancer cells whereby glucose is metabolized to produce lactate, even in the presence of oxygen [9]. Notably, during the Warburg effect, glucose uptake is accelerated by the upregulation of the glucose transporter 1 (GLUT1) and glycolytic enzymes. Lactate dehydrogenase A (LDHA) catalyzes the conversion of pyruvate to L-lactate and nicotinamide adenine dinucleotide (NADH) to NAD^+^ [10]. Lactate is exported from the cells via a monocarboxylate transporter (MCT), resulting in its accumulation in the microenvironment. The induction of glycolysis and the accumulation of lactate promote cell proliferation, progression, metastasis, and immune evasion, enhancing cancer progression [11,12,13].

A decreased mitochondrial copy number and mitochondrial metabolic abnormalities are frequently observed in OSCC [14]. However, the molecular mechanisms underlying these changes are poorly understood. On the other hand, when EBV infects nasopharyngeal epithelial cells, mitochondrial abnormalities and associated metabolic disorders have been observed [15], followed by the induction of epithelial to mesenchymal transition (EMT) and an increase in stem cells [16]. Our previous study has shown that infection of squamous epithelial cells with EBV induced EMT [6]. However, how EBV infection in OSCC affects changes in the mitochondrial number and metabolism remains largely unstudied. Moreover, the pathogenic role of EBV in OSCC progression, particularly in the regulation of glucose metabolism, has not yet been elucidated.

In this study, we attempted to clarify the role of EBV in OSCC progression through the induction of the Warburg effect. We investigated whether EBV infection upregulated genes associated with the Warburg effect and induced lactate production. Moreover, we investigated whether the upregulation of glycolytic enzymes correlated with tumor progression using a xenograft model. Our results may help identify therapeutic targets for EBV-associated OSCC by providing new insights into how EBV contributes to OSCC progression.

## 2. Results

### 2.1. EBV Induces Mitochondrial Stress by Reducing mtDNA Copy Number

It has been shown that EBV infection reduces mitochondrial biogenesis and mtDNA in EBV-infected B cells and monocytes [17,18]. However, little is known about whether EBV infection contributes to OSCC progression by inducing mitochondrial stress.

The mtDNA copy number quantified by qPCR showed a more than 25% decrease in EBV-positive cells compared with that in EBV-negative cells (Figure 1a). Immunofluorescence staining showed that the expression of mitochondrial HSP60 was significantly reduced in EBV-positive cells compared to that in parental EBV-negative cells (Figure 1b,c). These results showed that EBV infection reduced the mtDNA copy number. Thus, it was necessary to investigate whether the mitochondrial stress induced by EBV infection impaired mitochondrial function.

### 2.2. EBV Infection Mediates Metabolic Reprogramming by Inducing the Warburg Effect

The main role of mitochondria is to supply energy to the cells by producing adenosine triphosphate (ATP) via oxidative phosphorylation [19]. Therefore, mitochondrial stress may affect cellular metabolism. Darekar et al., showed that EBV infection of B lymphocytes induces the Warburg effect, which subsequently promotes cancer progression [20]. However, induction of the Warburg effect by EBV infection in OSCC has not been investigated. Our transcriptome analysis demonstrated that the EBV infection modulates the expression of genes by upregulation of Warburg effect-associated genes. However, the genes that were associated with the TCA cycle and electron transport chain were obviously downregulated in EBV-positive cell lines (Appendix A). Therefore, we analyzed the genes related to the Warburg effect using RT-qPCR. The expression levels of *LDHA*, *LDHB*, *GLUT1*, and phosphoinositide-dependent kinase-1 (*PDK-1*) were significantly higher in SCC25-EBV cells than in SCC25 cells (Figure 2a–d). In contrast, only *GLUT1*, but not *LDHA* and *LDHB*, was upregulated in HSC1-EBV cells compared with HSC1 cells (Figure 2a–d). Moreover, the immunofluorescence assay showed that the expression of GLUT1 was higher in EBV-positive cells than in EBV-negative cells in both HSC1 and SCC25 cells (Figure 2h). Furthermore, the concentration of lactate in EBV-positive cells was higher than in EBV-negative cells, and lactate accumulated based on the length of culture (Figure 2e,f). Consistent with these findings, LDH activity also increased in EBV-infected cells (Figure 2g). These results demonstrate that EBV mediated metabolic reprogramming in both HSC1 and SCC25 cells by inducing glucose metabolism via the Warburg effect.

### 2.3. EBV Infection Enhances Cancer Stem Cell Marker Expression and Tumor Initiation in OSCC Cells

The Warburg effect has been reported to be associated with cancer stem cells (CSCs) that promote tumor growth [21]. Furthermore, in EBV-associated epithelial malignancies, EBV promotes tumor growth by inducing CSCs [22,23]. Therefore, we measured the expression of CSC markers such as *CD44*, *CD44v6*, *CD44v9*, and *CD133* by RT-qPCR in OSCC cell lines. EBV infection significantly upregulated the expression levels of *CD44*, *CD44v6*, *CD44v9*, and *CD133* in SCC25-EBV cells, but not in HSC1-EBV cells (Figure 3a–d). Immunohistochemistry showed that the CD44 expression level was higher in SCC25-EBV cells than that in the parental SCC25 cells (Figure 3e). However, the CD44 expression level was lower in HSC1-EBV cells than that in the parental HSC1 cells (Figure 3e). These results indicate that EBV infection induced the expression of CSC marker proteins in a cell type-specific manner.

### 2.4. EBV Promotes Tumor Formation and Growth an In Vivo Mouse Model

We have previously shown that EBV infection promotes OSCC progression by inducing EMT, cell migration, cell invasion, and inhibition of apoptosis [6]. Furthermore, we showed that EBV infection induced the Warburg effect using an in vitro cell culture model (Figure 2). Therefore, we hypothesized that EBV infection might promote OSCC progression via the Warburg effect. To test this hypothesis, we established an experimental mouse xenograft model.

SCC25 and SCC25-EBV cells were injected subcutaneously into SCID mice and the sizes of the developed tumors were measured. After 14 weeks, the tumor volume induced by SCC25-EBV cells was significantly greater than that induced by SCC25 cells (Figure 4a). A histological examination revealed that tumors formed by SCC25-EBV cells exhibited a more proliferative and undifferentiated phenotype than those formed by SCC25 cells (Figure 4b,c). These results support our hypothesis that EBV infection promotes tumor progression in OSCC via the Warburg effect.

### 2.5. EBV Promotes Tumor Formation in OSCC after Passage by Xenograft Model

We further clarified the role of EBV in promoting OSCC progression by investigating tumor cell migration and invasion. When SCC25-EBV tumor cells were serially passaged in mice, their ability to migrate and invade significantly increased compared to their parental cells. However, no such increase was observed in the EBV-negative SCC25 cells (Appendix A). The gene expression levels of *LDHA* and *GLUT1* were examined in SCC25-EBV tumor cells to investigate whether the malignant phenotypes of OSCC were induced by the Warburg effect. The mRNA levels of *LDHA* and *GLUT1* were significantly upregulated in SCC25-EBV cells passaged in mice compared to those in parental cells (Figure 5a,b). GLUT1 staining in SCC25-EBV tumor cells showed an increased intensity and number of positive cells (Figure 5c). Similarly, the level of lactate released into the culture supernatant was higher in SCC25-EBV cells passaged in mice than in the parental cells (Figure 5d). These results suggest that EBV infection promoted OSCC progression by inducing the Warburg effect.

The expression levels of CSC markers were measured using RT-qPCR. *CD44* was significantly upregulated in SCC25-EBV cells passaged in mice compared to parental cells (Figure 6a). However, the *CD44* expression did not increase in SCC25 cells passaged in mice. The expression of *CD44v6* and *CD44v9* was significantly upregulated in SCC25-EBV cells passaged in mice. In addition, expression in second-passage tumor cells was higher than that in the initial tumor cells. However, the expression levels of *CD44v6* and *CD44v9* decreased in SCC25 cells passaged in mice (Figure 6b,c). Consistent with the mRNA levels, CD44 protein levels were dramatically increased in SCC25-EBV cells, especially in cells passaged in mice (Figure 6d). These results indicate that EBV promoted OSCC progression by inducing CSC properties and the Warburg effect.

### 2.6. EBV Increases the Number of Tumor-Initiating Cells In Vivo

To clarify the tumor-initiating activity of EBV-infected cells, cancer cells isolated from tumor tissues were serially passaged into BALB/c and SCID mice. The number of cells injected into the BALB/c mice decreased by a factor of five with each inoculation (1 × 10^6^ or 2 × 10^5^ cells for the second and third injections, respectively). However, SCC25-EBV tumors grew more rapidly after the third injection than after the first or second injection (Figure 7a). In another experiment using SCID mice, we reduced the number of SCC25-EBV cells injected into the mice by a factor of 10 at each passage (1 × 10^6^ or 1 × 10^5^ cells for the second and third injections, respectively). Similarly, the tumors from the third injection were significantly larger than those from the first and second injections (Figure 7b). A histological analysis showed that SCC25-EBV tumor cells from the third injection exhibited a well-differentiated but highly malignant appearance compared to tumors from the first and second injections (Appendix A). In contrast, EBV-negative SCC25 cells did not pass through the xenograft model and disappeared 4 weeks after injection in both BALB/c and SCID mice. These results suggest that EBV infection promotes OSCC progression by stimulating OSCC tumor cell growth.

## 3. Discussion

Infection with several human oncogenic viruses, such as human papillomavirus and EBV, affects mitochondrial function [24,25]. In normal cells, mitochondrial function is important for metabolism, immune responses, cell death, and homeostasis [26,27]. In contrast, in cancer cells, mitochondrial metabolic reprogramming plays a crucial role in cell proliferation as it provides energy to the cells [28]. The metabolic remodeling of EBV-infected cells has been demonstrated during B-lymphocyte transformation [29]. However, the metabolic changes associated with EBV infection have only been reported in nasopharyngeal epithelial cells [15,16]. Here, we showed that EBV infection of OSCC cells reduced the mtDNA copy number (Figure 1). Moreover, EBV infection of OSCC cells induced metabolic reprogramming of OSCC cells and upregulated GLUT1 and LDHA expression levels, demonstrating the Warburg effect (Figure 2).

Expression of EBV genes in infected cells has been reported to affect mitochondrial function. The viral immediate-early gene, *BamHI* Z fragment leftward open reading frame-1 (BZLF1), interacts with mitochondrial single-stranded DNA-binding proteins [17]. Furthermore, BZLF1 expression alters mitochondrial morphology and reduces the mtDNA copy number in B cells [30]. The EBV latent membrane protein 2A (LMP2A) also alters mitochondrial dynamics by inducing the expression of Drp1, which is important for mitochondrial fission [31].

EBV induces metabolic reprogramming in chronically infected cells. Long-term EBV infection in B cells reduces mitochondrial respiration by downregulating the *tricarboxylic acid* (TCA) cycle and oxidative phosphorylation-related genes through enhanced glucose import and GLUT1 expression [32]. In addition, LMP1 alters cell metabolism by affecting a range of processes from mitochondrial respiration to aerobic glycolysis by inducing hypoxia inducing factor-1α (HIF-1α) through the activation of poly [ADP-ribose] polymerase 1 [33]. In lymphoblastoid cell lines, EBV infection induces GLUT1 expression and glucose metabolism via the IKKβ/NF-κB signaling pathway [34]. Furthermore, EBV infection of B cells promotes the Warburg effect by inducing MYC and HIF-1α expressions [35,36]. In NPC cells, EBV infection may induce cell migration and invasion through the Warburg effect because LMP1 induces glucose metabolism by activating the IGF1/mTORC2/AKT and mTORC1/NF-κB signaling pathways [37,38]. Moreover, EBV miR-BART1-5p induces lactate production and glucose consumption by inducing GLUT1, Hexokinase 2, LDHA, and HIF-1α expressions through the activation of the 5′ AMP-activated protein kinase (AMPK)/mTOR/HIF-1α pathway that enhances angiogenesis and cell proliferation [39].

The promotion of aerobic glycolysis and suppression of reactive oxygen species (ROS) production have been reported to increase CSC-like properties and tumorigenicity [40]. CSCs regulate the ability to induce tumor initiation and maintenance [41]. Additionally, CD44 and its variants (CD44v6 and CD44v9), CD133, CD24, epithelial cell adhesion molecule (EpCAM), leucine-rich repeat-containing G-protein coupled receptor 5 (LGR5), and ATP-binding cassette super-family G member 2 (ABCG2), have been recognized as CSC markers [42,43]. We showed that EBV infection promoted the stemness of OSCC cells by inducing the expression of CSC markers CD44, CD44v6, CD44v9, and CD133 (Figure 3). In addition, the xenograft models showed that EBV infection induced OSCC tumorigenesis and promoted tumor growth (Figure 4). The EBV-induced malignant phenotype in OSCC correlated with the upregulation of GLUT1 and LDHA and increased lactate production and LDH activity (Figure 5 and Figure 7).

The mechanism by which EBV infection promotes stemness in epithelial cancers other than EBV-associated OSCC has been investigated. Infection of NPC cells with EBV can increase the CD44-positive population, which shows enhanced tumorigenicity and chemoresistance compared with the CD44-negative population [44,45]. EBV EBNA1, LMP1, and LM2A play important roles in expressing stem cell phenotypes in NPC cells by inducing sonic hedgehog (SHH) ligands and Hh pathway activation [46]. LMP1 also upregulates CSC markers such as CD44 and ATP-binding cassette superfamily G member 2 (ABCG2), and increases the number of side population cells, self-renewal properties, and the tumorigenicity of NPC cells through the PI3K/AKT signaling pathway [47,48,49]. LMP1 also increases CSC properties in NPC cells [50]. Among the EBVaGC cells, sphere-forming cells (SFCs) isolated from SNU-719 cells express CD44 at high levels and express CD24 at low levels. However, this is not observed with CD133, CD338, or CD90. These SFCs exhibit CSC-like properties and increased tumorigenicity in a mouse xenograft model [51]. In addition, the ectopic expression of LMP2A contributes to CSCs in EBVaGC cells by inducing in vitro spheroid formation and tumor-initiating ability in vivo [23].

Our results revealed that EBV infection in OSCC cells induced mitochondrial stress by reducing the mtDNA copy number. Genes exhibiting the Warburg effect, including *LDHA*, *GLUT1*, and *PDK1,* were upregulated in EBV-positive SCC25 cells, which also increased lactate production and LDH activity in EBV-positive SCC25 cells. In a mouse transplantation model, the expression levels of cancer stem cell markers, such as CD44 and CD133, were upregulated (Figure 6), and tumor growth was promoted (Figure 4a) in SCC25 cells by EBV infection. Furthermore, the tumorigenicity of EBV-infected SCC25 cells increased with the number of passages (Figure 7). In contrast, HSC1 cells differed from SCC25 cells. Although EBV infection of HSC1 cells induced mitochondrial stress (Figure 1), it did not upregulate the expression levels of PDK1, CD44, CD44v6, CD44v9, or CD133 in these cells (Figure 2 and Figure 3). Our results suggest that EBV infection promotes CSC-like properties by inducing the Warburg effect during OSCC progression.

HSC1-EBV cells transiently formed tumors in SCID mice; however, all tumors regressed and could not be observed after 4 weeks, and only HSC1 cells formed tumors (Appendix A). Our previous study demonstrated that HSC-EBV exhibited a strong expression of LMP1 and induced the lytic replication [6]. LMP1 plays a pivotal role in the induction of lytic replication of EBV by inducing an Rta promoter [51]. In addition, LMP1 can simultaneously induce and inhibit apoptosis in B cells [52]. Furthermore, our transcriptomics revealed that EBV infection of HSC1 cells could alter the expression pattern of host genes by, at least, the upregulation of genes that were associated with the apoptotic process, positive regulation of the apoptotic process, and the intrinsic apoptotic signaling pathway in response to DNA damage (Appendix A.) and activated the TNF-α signaling pathway through the activation of the NF-κB signaling pathway [6]. Therefore, a strong expression of LMP1 may lead to a high immunogenicity, induce lytic replication of EBV and, thus, causes the failure of tumor formation.

## 4. Materials and Methods

### 4.1. Cell Lines

EBV-negative HSC1 and SCC25 cells, as well as EBV-positive HSC1-EBV and SCC25-EBV cells [6], were cultured in Dulbecco’s Modified Eagle Medium (DMEM)/Ham’s F-12 (Wako, Osaka, Japan) supplemented with 10% fetal bovine serum (FBS) (HyClone^TM^, GE Healthcare Life Sciences, Logan, UT, USA), 100 U/mL penicillin, and 100 μg/mL streptomycin (Nacalai Tesque Inc., Kyoto, Japan). The cells were cultured at 37 °C in a humidified incubator with 5% CO_2_.

### 4.2. Gene Expression Analysis Using Reverse Transcription Quantitative-Polymerase Chain Reaction (RT-qPCR)

Total RNA was extracted from cells using the ISOGEN reagent (Nippon Gene, Tokyo, Japan) as per the manufacturer’s instructions. Then, 1 µg of RNA was used to synthesize complementary DNA using SuperScript^®^ III Reverse Transcriptase (Invitrogen, Carlsbad, CA, USA). Gene expression was quantified by RT-qPCR using SsoAdvanced^TM^ SYBR^®^ Green Supermix on a CFX Connect Real-Time PCR System (Bio-Rad Laboratories, Hercules, CA, USA). Glyceraldehyde 3-phosphate dehydrogenase (*GAPDH*) was used as an internal control. Relative quantitative expression levels of analyzed genes were determined using the 2^−ΔCT^ method. The primers used in this study are listed in Table 1.

### 4.3. Quantification of Mitochondrial DNA (mtDNA) Copy Number

Genomic DNA was extracted from cells using GenElute™ Mammalian Genomic DNA Miniprep Kit (Sigma-Aldrich, St. Louis, MO, USA) according to the manufacturer’s instructions. DNA samples were amplified by PCR using tRNA and B2M primers. All assays were performed in duplicate on the qPCR instrument using SsoAdvanced^TM^ SYBR^®^ Green Supermix. The mtDNA copy number was calculated according to a previous study [52]. tRNA was used as an internal control.

### 4.4. Immunofluorescence Staining

Cells were grown on sterilized 12-holes microslide glass (Matsunami, Osaka, Japan) at 1.5 × 10^3^ cells/hole overnight. Cells were washed twice with calcium free PBS (PBS-) and probed with primary antibodies against heat shock protein 60 (HSP60) (D6F1, Cell Signaling Technology, Beverly, MA, USA) and GLUT1 (CSB-PA002728, Cusabio Biotech, College Park, MD, USA) at 4 °C for 30 min. Cells were subsequently stained with goat anti-rabbit IgG Hilyte Fluor 647 (AnaSpec, Fremont, CA, USA). CD44 was stained by PE conjugated antibody (IM7, BioLegend, San Diego, CA, USA). Additionally, 4′,6-diamidino-2-phenylindole (DAPI) (Wako) was used for staining the nucleus. Expression and localization of proteins were examined using an FV1000D laser scanning confocal microscope (Olympus, Tokyo, Japan).

### 4.5. Lactate Production and Lactate Dehydrogenase (LDH) Activity Analyses

Cells were seeded in 96-well plates at a density of 5 × 10^3^ cells/well and incubated for 24, 48, 72, and 96 h. The concentration of lactate in the cell culture supernatants at each time point was determined using a lactate assay kit-WST (DOJINDO, Kumamoto, Japan) according to the manufacturer’s instructions. Additionally, LDH activity was determined using a cytotoxicity LDH assay kit-WST (DOJINDO), according to the manufacturer’s instructions.

### 4.6. Cell Migration and Invasion Analysis Using the Transwell Assay

Cell migration and invasion assays were performed as previously described [6] using Transwell chambers (BD Biosciences, San Jose, CA, USA) and 0.8 µm Costar^®^ polycarbonate membrane Transwell^®^ inserts (Costar, Croning, NY, USA). Each group was assayed in triplicates.

### 4.7. In Vivo Study

Female severe combined immunodeficiency (SCID) mice and BALB/c nude mice (5 weeks of age) (Charles River Laboratories, Wilmington, MA, USA) were used to set up an in vivo model to investigate whether EBV infection promoted tumor formation and growth. Cells were subcutaneously injected into both flanks of SCID mice at a density of 1 × 10^7^ cells. Mice were randomly divided into five groups (*n* = 3/group): a negative control group injected with PBS (group 1), SCC25 cells (group 2), and SCC25 green fluorescent protein (GFP)/EBV cells (group 3). After 11–16 weeks, the mice were anesthetized by intraperitoneal injection of a mixture containing 0.75 mg/kg medetomidine (Orion, Espoo, Finland), 4 mg/kg midazolam (Sandoz, Holzkirchen, Germany), and 5 mg/kg butorphanol (Bristol-Myers Squibb, New York, NY, USA) and the tumors were excised. Tumor size was measured weekly using calipers and body weight was monitored every 3 days. The tumor volume based on caliper measurements was calculated using the following formula:$$Tumor\;volume=\frac{1}{2}(length\times {\left(weight\right)}^{2})$$

Animal experiments were conducted under the supervision of Shimane University, Faculty of Medicine, and the experimental animal program was approved by the Animal Research Committee.

### 4.8. Analysis of OSCC Stemness Induced by EBV Infection

Cells isolated from tumor tissues were treated with an enzyme cocktail containing 0.4 mg/mL collagenase (Gibco, Waltham, MA, USA) and 0.2 mg/mL dispase (Gibco, Waltham, MA, USA) in RPMI-1640 (Sigma-Aldrich), then incubated at 37 °C for 4 h. Isolated cells were washed and cultured in DMEM/F-12 (Wako) containing 10% FBS (GE Healthcare Life Sciences), 1% penicillin/streptomycin (Nacalai Tesque Inc.) and 1% anti-Mycoplasma (BIOMYC-3, Nacalai Tesque Inc.). Cells were subcutaneously injected into both flanks of BALB/c nude and SCID mice at a 10–fold lower number than that at the first injection. The mice were euthanized after 9 weeks injection. Tumor size was measured weekly using calipers, and body weight was monitored every 3 days. Cells isolated from the tumor tissues were subjected to further analysis.

### 4.9. Statistical Analysis

The GraphPad Prism software 7.0 (GraphPad Software Inc., San Diego, CA, USA) was used for all data analyses. The Whitney test was used to analyze whether there was a difference between two independent groups, which was expressed as mean ± standard deviation (SD). All experiments were repeated thrice. A *p* value < 0.05 was considered statistically significant.

## 5. Conclusions

We demonstrated that in EBV-associated OSCC, EBV infection affects mitochondrial activity by reducing the number of mtDNA copy, which subsequently reprograms the cellular metabolism to exhibit the Warburg effect by inducing lactate production through the upregulation of GLUT1 and LDHA. Induction of the Warburg effect driven by EBV infection promotes stemness and tumorigenesis in OSCC cells by upregulating CSC markers such as CD44, CD44v6, and CD44v9. These findings may provide new insights into how EBV contributes to OSCC progression and strategies for treating EBV-associated OSCC.

## Figures and Tables

**Figure 1 ijms-24-14072-f001:**
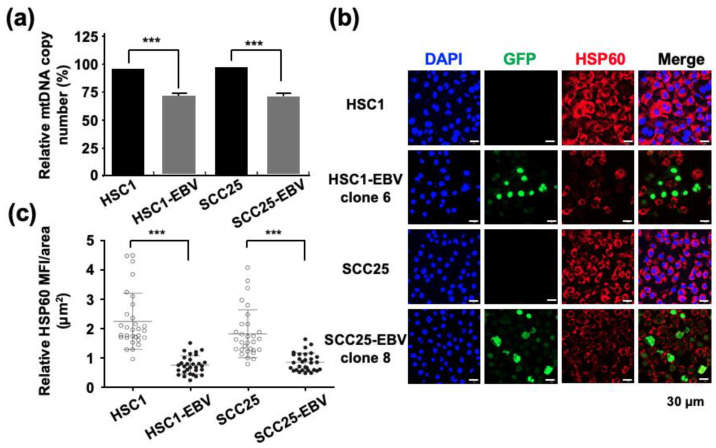
EBV infection decreases the mitochondrial copy number of infected cells. (**a**): the mtDNA copy number was examined by qPCR in EBV-positive and EBV-negative cells. (**b**): Mitochondrially localized HSP60 was stained in Red. Cells infected with recombinant GFP-EBV are shown in green. Nuclei were stained blue. (**c**): the strength of HSP60 expression was shown by mean fluorescence intensity. ***: *p* < 0.001. Scale bar: 30 µm.

**Figure 2 ijms-24-14072-f002:**
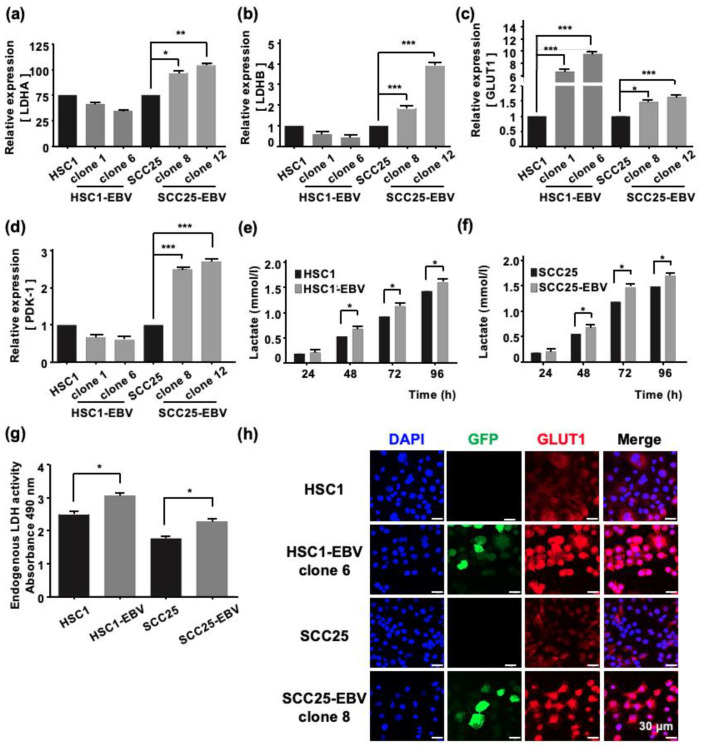
EBV infection of cells induces the Warburg effect. The expression levels of genes related with the Warburg effect, such as *LDHA* (**a**), *LDHB* (**b**), *GLUT1* (**c**), and *PDK-1* (**d**), were analyzed by RT-qPCR. The concentration of Lactate in HSC1 cells and HSC1-EBV cells (**e**) and SCC25 cells and SCC25-EBV cells (**f**) are shown. LDH activity of the different groups (**g**). The expression and localization of GLUT1 protein in HSC1, HSC1-EBV, SCC25, and SCC-EBV cells are shown (**h**). *: *p* < 0.05, **: *p* < 0.01, ***: *p* < 0.001. Scale bar: 30 µm.

**Figure 3 ijms-24-14072-f003:**
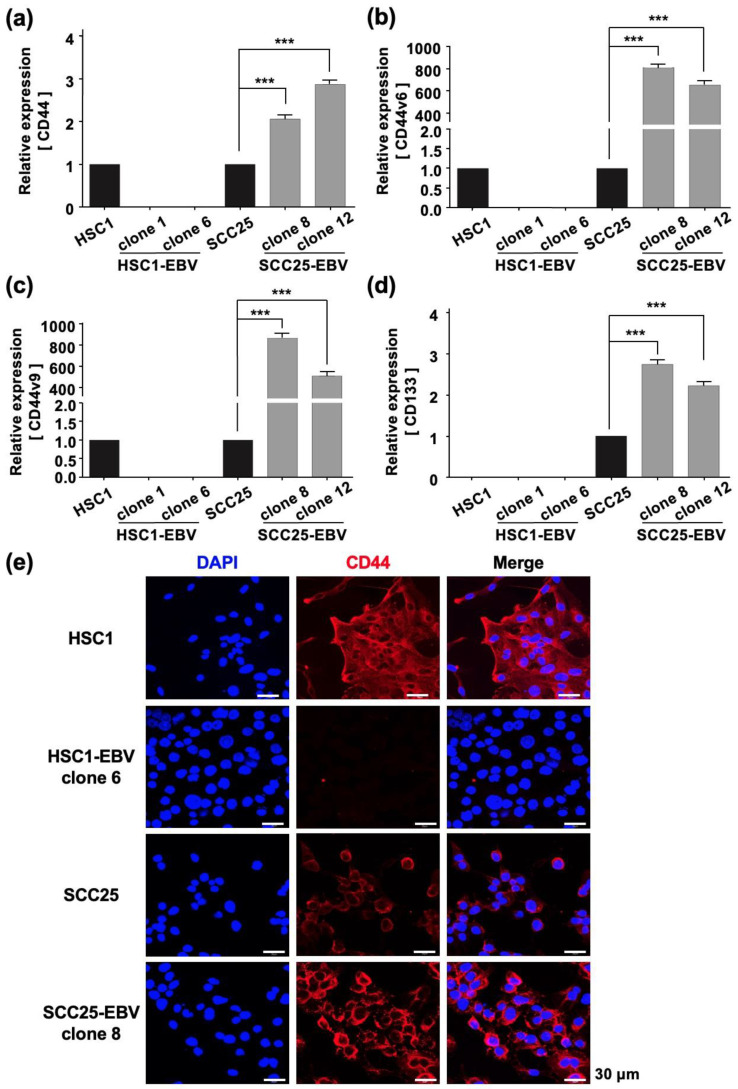
EBV infection induces stemness in SCC25 cells. Expression levels of cancer stem cell markers including CD44 (**a**), CD44v6 (**b**), CD44v9 (**c**), and CD133 (**d**) were investigated by RT-qPCR. CD44 protein expression and localization as investigated by immunofluorescence assay are shown (**e**). ***: *p* < 0.001. Scale bar: 30 µm.

**Figure 4 ijms-24-14072-f004:**
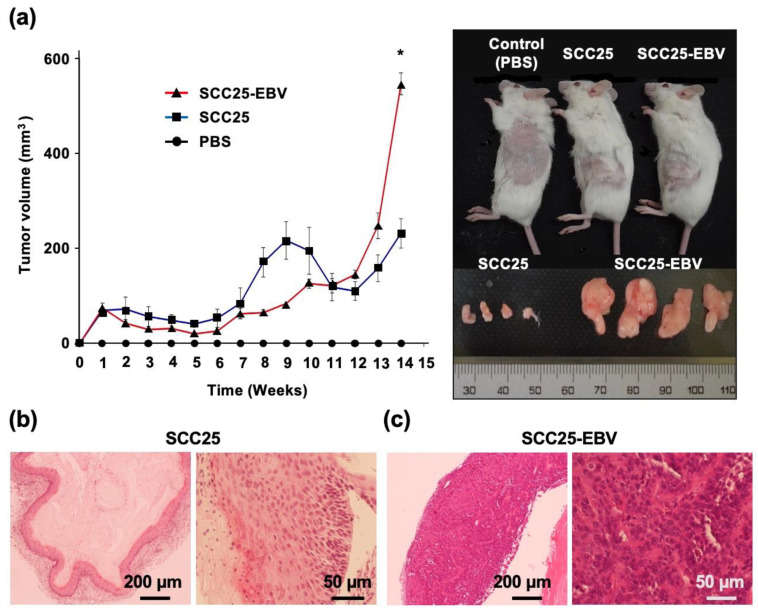
EBV infection induces tumor growth of OSCC in the xenograft model. (**a**): Left graph indicates tumor volumes of SCC25 and SCC25-EBV cells injected subcutaneously into SCID mice. Right figure shows tumors grown at the flank of the mice and the excised tumor. *: *p* < 0.05. (**b**,**c**): H&E staining of tumor tissues of SCC25 cells (**b**) and SCC25-EBV cells (**c**). Magification; 4× and 20×. Scale bar: 200 and 50 µm.

**Figure 5 ijms-24-14072-f005:**
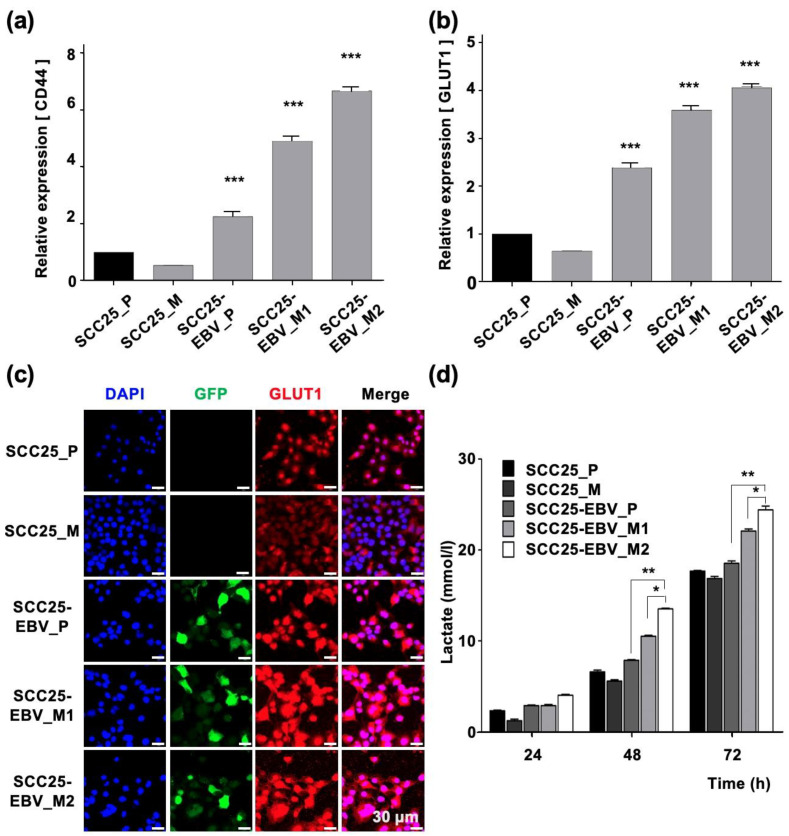
Passaging tumor cells in mice promotes the Warburg effect in EBV-positive cells, but not in EBV-negative cells. The mRNA levels of LDHA (**a**) and GLUT1 (**b**) were measured by RT-qPCR. The lactate production in parental cells and tumor cells passaged in mice was measured (**c**). The expression and localization of GLUT1 were examined by immunofluorescence assay (**d**). SCC25_P: SCC25 parental cells, SCC25_M: SCC5 cells isolated from mice, SCC25-EBV_P: SCC25-EBV parental cells, SCC25-EBV_M: SCC25-EBV cells isolated from mice and SCC25-EBV_M2nd: second injection of SCC25-EBV cells isolated from mice. *: *p* < 0.05, **: *p* < 0.01, ***: *p* < 0.001. Scale bar: 30 µm.

**Figure 6 ijms-24-14072-f006:**
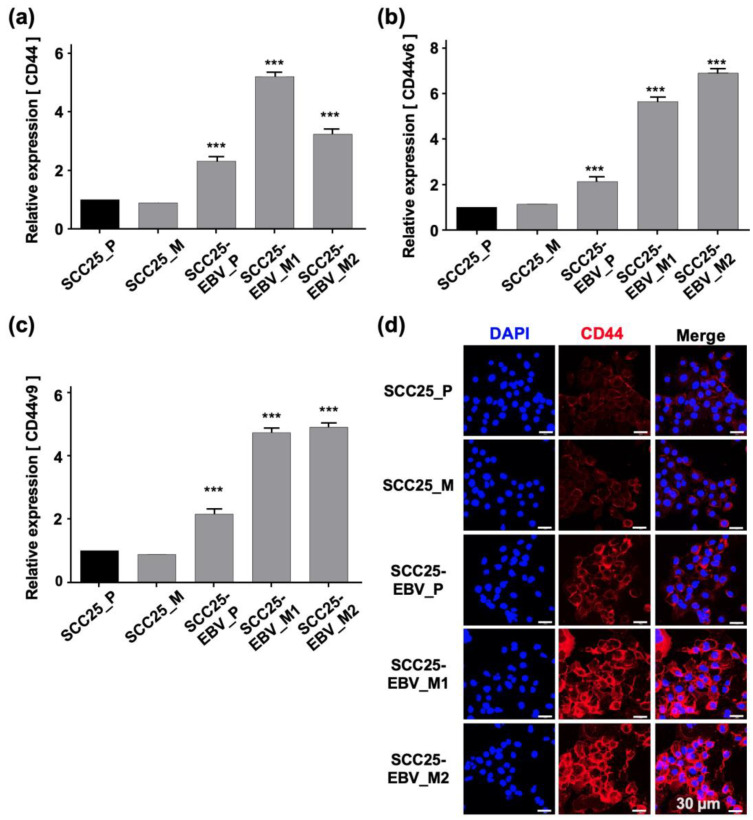
Promotion of stemness in OSCC cells by sequential passage in mice. The expressions of cancer stem cell markers, such as CD44 (**a**), CD44v6 (**b**), and CD44v9 (**c**), were examined by RT-qPCR. The expression level of CD44 protein was analyzed by immunofluorescence assay (**d**). SCC25_P: SCC25 parental cells, SCC25_M: SCC5 cells isolated from mice, SCC25-EBV_P: SCC25-EBV parental cells, SCC25-EBV_M: SCC25-EBV cells isolated from mice and SCC25-EBV_M2nd: second injection of SCC25-EBV cells isolated from mice. ***: *p* < 0.001. Scale bar: 30 µm.

**Figure 7 ijms-24-14072-f007:**
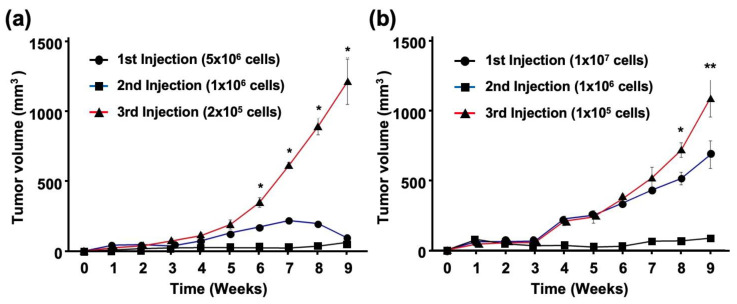
EBV promotes growth of OSCC tumor cells. SCC25-EBV cells were subcutaneously injected into the flank of either BALB/c (**a**) or SCID mice (**b**). *: *p* < 0.05., **: *p* < 0.01.

**Table 1 ijms-24-14072-t001:** Primer sequences.

Gene	Forward (5′-3′)	Reverse (5′-3′)
*B2M*	TGCTGTCTCCATGTTTGATGTATCT	TCTCTGCTCCCCACCTCTAAGT
*CD44-F*	CAACCGTTGGAAACATAACC	CAAGTGGGAACTGGAACGAT
*CD44v6*	CCAGGCAACTCCTAGTAGTACAACG	CGAATGGGAGTCTTCTTTGGGT
*CD44v9*	GGC TTG GAA GAA GAT AAA GAC C	TGCTTGATGTCAGAGTAGAAGTTG
*CD44v9*	GGC TTG GAA GAA GAT AAA GAC C	TGCTTGATGTCAGAGTAGAAGTTG
*GAPDH*	AGCCACATCGCTCAGACAC	GCCCAATACGACCAAATCC
*GLUT1*	ACTGGGCAAGTCCTTTGAGAT	GTCCTTGTTGCCCATGATGGA
*HIF-1a*	GTGTACCCAACTAGCCGAG	GCACTGTGGTTGAGAATTCTTGG
*LDHA*	GGCCTGTGCCATCAGTATCT	CTTTCTCCCTCTTGCTGACG
*LDHB*	GGACAAGTTGGTATGGCTGTG	AAGCTCCCATGCTGCAGATCCA
*PDK-1*	CTGTGATACGGATCAGAAACCG	TCCACCAAACAATAAAGAGTGCT
*PROM1*	AGTCGGAAACTGGCAGATAGC	GGTAGTGTTGTACTGGGCCAAT
*tRNA*	CACCCAAGAACAGGGTTTGT	TGGCCATGGGTATGTTGTTA

## Data Availability

The published articles included all datasets generated or analyzed during this study.

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
