# Peer review of "Epstein–Barr Virus Promotes Oral Squamous Cell Carcinoma Stemness through the Warburg Effect"

_ijms, 2023, doi:10.3390/ijms241814072_

Round 1

Reviewer 1 Report

An overall nice and well-reported study

Introduction and abstract are adequate

Methods well explained

Results properly described

Discussion and refernces adequate.

I have only one concern. The authors somehow claim that the increased aggressiveness of EBV+ cases is mediated by the observed metabolic change. However, they do not explore anything else. The message, therefore, must be mitigated. The clinical aggressiveness is accompained by wrburg effect and so on. But there is no evidence that this is the driving cause.

Author Response

Response:  We thank the reviewer for the valuable comments.

It has been reported that the infection of EBV, by expressing LMP1, mediates the Warburg effect in nasopharyngeal cancer (NPC) cell lines through the activation of the IGF1/mTORC2/AKT and mTORC1/NF-κB signaling pathways. In addition, the EBV-microRNA, EBV miR-BART1-5p, also induces lactate production and glucose consumption by inducing GLUT1, Hexokinase 2, LDHA, and HIF-1α expressions through the activation of 5' AMP-activated protein kinase (AMPK)/mTOR/HIF-1α pathway. The activation of these signaling pathways could further promote tumor cell aggressiveness [Revision references 1-3].

In line with this, we have shown in a previous study that EBV-positive squamous cells express LMP1 and LMP2A, which can activate the Warburg effect and promote tumor progression [Revision reference 4]. Similar to the expression of LMP1 and LMP2A, EBV-positive squamous cells also show a distinct increase in the expression of exon 4 of EBV-microRNAs (Revision Figure 1). Furthermore, our transcriptome analysis also revealed that EBV infection upregulates the expression of genes associated with the Warburg effect, including GLUT1, LDHA, and PDK-1. However, the expression of genes related to the TCA cycle and electron transfer chain was downregulated in EBV-positive cell lines (Figure S4). These results thus indicate that EBV infection promotes the tumorigenic potential of OSCCs by inducing the Warburg effect at least through the function of LMP1, LMP2 and EBV-microRNAs.

We have modified the manuscript by adding "Our transcriptome analysis demonstrated that EBV infection modulates gene expression through the upregulation of genes related to the Warburg effect. However, genes associated with the TCA cycle and electron transfer system were clearly downregulated in EBV-positive cell lines (Figure S4)." Please see lines from 101 to 104 in the revised manuscript and Figure S4.

Revision Figure 1. The expression of EBV genes. The expression of latent proteins (A) by western blotting. The expression of EBER1 and EBER2 (B) LMP1, LMP2A, and EBNA1 (C) and EBV-microRNA, BART exon4, (D) in EBV positive cell lines by qRT-PCR.

Figure S4. The expression levels of genes that were associated with glycolysis (A), TCA cycle (B) and electron transport chain (C) in EBV-positive and EBV-negative cell line quantified by microarray technique.

Reviewer 2 Report

The study (by Heawchaiyaphum et al.) focuses on the human oncogenic Epstein-Barr virus (EBV) and examined the mechanism utilized by EBV to drive oral squamous cell carcinoma (OSCC) cancer development. Importantly, EBV infection is associated with increased risk for OSCC, but exact molecular mechanisms are still unclear, hindering finding future therapeutic targets. Therefore, it is very important to better understand underlying events related to cancer progression and finding cancer-relevant mechanisms triggered by EBV.  The authors have previously utilized two independent, highly different OSCC cells lines, and generated and characterized their EBV Infected derivatives (namely SCC25, HSC1 and their EBV containing versions), which are also utilized in this study. They and others have already showed that EBV infection promotes OSCC progression using these cell lines. The 2 OSCC cell lines allow them to identify cell type specific and potentially more universal EBV-driven mechanisms as well.  

The impact of EBV on cellular metabolism and mitochondrial function has been shown in previous studies in other context (for example in B-cells), here the authors convincingly demonstrate the reduced mtRNA copy number and higher expression of metabolic marker genes by EBV infection in both OSCC EBV cell lines.   Notably, impact of EBV on OSCC cell lines are similar, but not identical, potentially highlighting cell type specificity and context specific impact on the host, in triggering cellular changes. For example, the results further highlight the role of EBV in inducing Warburg effect in the cells, and reveal the induction of only a small subset of metabolic genes in EBV-infected HSC1 cells, as opposed to all tested genes were found to be expressed at higher levels in SCC25 EBV cell lines compared to SCC25 cell line. Furthermore, in line with the same finding, the authors show higher cancer stem cell (CSC) gene expression in EBV positive OSCC, but only in the SCC25 EBV cell lines, but not in HSC1 EBV and concluding on potential cell type specificity.

The study also suggest that EBV infection promotes OSCC progression by Warburg effect., which was even more pronounced after serial passaging in mice.

Overall, the study is well executed, with good quality figures and datasets, and fills a critical knowledge gap. In summary, although not all findings are identical with both cell line models, but this can be also critical information as it might inform us about context specific impacts.  To further strengthen their conclusion and better reflect the findings, it would be good to refine the statements and clarify the indicated minor issues below:

1.     Fig 4A: Notably,. only 1 timewindow was found to be significant (at week 14, Fig 4A), which finding was not discussed in the manuscript. Please discuss and elaborate on why the EBV infected SCC25 OSCC cell lines promote tumor growth only after 11th week in the xenograft model, but not earlier. Importantly, only 1 single timepoint in this experiment supports the claim, yet all the other previous timepoints show opposite conclusion (but not a significant difference). this needs more elaboration for clarification. Please provide justification for the time choosen for conclusion. How did the authors decide to focus on the given timewindow (week 14) to end the experiment?

2.     Fig S3/Row 293: The authors discuss the fact that HSC1-EBV cells regress, but HSC1 did form tumors in mice. This finding directly contradicts the finding with SCC25-EBV vs, SCC15 findings. The authors should elaborate on potential reason, possible mechanisms for these findings, as this is currently at the very end of the discussion but lacks any elaboration. For example, is there any link between EBV latency (e.g. lytic viral gene expression)? Importantly, differences are also important to reveal, as it can further promote studies.  Authors should consider strengthening their findings with an additional OSCC EBV cell line for some of the in vitro experiments at least. As a minimum,  revise the statement to better highlight and reflect that only one of the model cell lines support the last findings in the tumor xenograft model, and the strong statements. 

Author Response

  1. Fig 4A: Notably, only 1 time window was found to be significant (at week 14, Fig 4A), which finding was not discussed in the manuscript. Please discuss and elaborate on why the EBV infected SCC25 OSCC cell lines promote tumor growth only after 11th week in the xenograft model, but not earlier. Importantly, only 1 single timepoint in this experiment supports the claim, yet all the other previous timepoints show opposite conclusion (but not a significant difference). This needs more elaboration for clarification. Please provide justification for the time chosen for conclusion. How did the authors decide to focus on the given time window (week 14) to end the experiment?

Response:  We thank the reviewer for the valuable comment. Our previous study demonstrated that the SCC25-EBV cells express only EBNA1 protein, but not other oncoproteins, such as LMP1 and LMP2A [Revision rference 1]. Therefore, we thought that EBV might not be able to promote rapid tumor cell growth.  Consistently, we have observed that EBV infection in SCC25 cells did not promote cell proliferation in vitro.

  The reasons why we did not keep mice no longer than 14 weeks are as follows;

  1. We were concerned about the effects of changes in the health status of mice. If mice are kept for more than 14 weeks, other factors such as immunity, age, and hormones could affect the experiment.
  2. Tumors formed by SCC25-EBV cells became larger than 500 mm3.
  3. We would like to determine the tumor-initiating activity of EBV-infected cells by passaging tumor cells to another mice.

Therefore, we decided to transfer tumors to the new mice as shown in Fig. 7.

  1. Fig S3/Row 293: The authors discuss the fact that HSC1-EBV cells regress, but HSC1 did form tumors in mice. This finding directly contradicts the finding with SCC25-EBV vs, SCC15 findings. The authors should elaborate on potential reason, possible mechanisms for these findings, as this is currently at the very end of the discussion but lacks any elaboration. For example, is there any link between EBV latency (e.g. lytic viral gene expression)? Importantly, differences are also important to reveal, as it can further promote studies. Authors should consider strengthening their findings with an additional OSCC EBV cell line for some of the in vitro experiments at least. As a minimum, revise the statement to better highlight and reflect that only one of the model cell lines support the last findings in the tumor xenograft model, and the strong statements.

Response: We thank the reviewers for bringing up some very important points. Our previous study (Microorganisms 2020, 8, 419: Revision reference 1) revealed the expression pattern of EBV genes between HSC1-EBV and SCC25-EBV is different (Figure 2). Both HSC1-EBV cells and SCC25-EBV cells expressed EBNA1, EBER1, and EBER2 (Figure 2C, D). However, LMP1 and LMP2A proteins were detected in HSC1-EBV cells, but not SCC25-EBV cells (Figure 2C) [Revision reference 1].

In addition, LMP1 is abundantly expressed during the lytic cycle of viral replication [Revision references 2, 3].

Consistently, we previously observed that HSC-EBV cells spontaneously produced a high viral titer, but SCC25-EBV cells did not produce [Revision reference 1]. Previous study demonstrated that LMP1 can activate apoptosis [Revision reference 4]. In addition, our transcriptome analysis demonstrated that EBV infection in HSC1 cells up-regulated the expression of genes that are associated with apoptotic process, positive regulation of apoptotic process and intrinsic apoptotic signaling pathway in response to DNA damage  [Revision reference 5]. Moreover, EBV infection in HSC1 cells activated the TNF-α signaling pathway through the activation of NF-kB signaling pathway [Revision reference 5].

Taken together, a strong expression of LMP1 may lead to a high immunogenicity, gene expression pattern alteration, and induce lytic replication of EBV, thus, causes the apoptosis of HSC1-EBV cells. Please see lines from 300 to 310 in the revised manuscript and Supplementary Table 1.

Supplementary Table 1. The significantly up-regulated apoptosis-associated genes in HSC1-EBV cells.

Entrez gene ID

Gene name

Log2(FC)

9274

BAF chromatin remodeling complex subunit BCL7C (BCL7C)

4.31

581

BCL2 associated X, apoptosis regulator (BAX)

3.92

27113

BCL2 binding component 3 (BBC3)

4.07

666

BCL2 family apoptosis regulator BOK (BOK)

3.01

598

BCL2 like 1 (BCL2L1)

3.20

83596

BCL2 like 12 (BCL2L12)

2.10

440603

BCL2 like 15 (BCL2L15)

3.03

8837

CASP8 and FADD like apoptosis regulator (CFLAR)

3.33

1048

CEA cell adhesion molecule 5 (CEACAM5)

4.95

4680

CEA cell adhesion molecule 6 (CEACAM6)

5.60

10395

DLC1 Rho GTPase activating protein (DLC1)

2.60

220042

DNA damage induced apoptosis suppressor (DDIAS)

2.41

55332

DNA damage regulated autophagy modulator 1 (DRAM1)

2.23

54097

FAM3 metabolism regulating signaling molecule B (FAM3B)

2.02

23017

Fas apoptotic inhibitory molecule 2 (FAIM2)

2.02

355

Fas cell surface death receptor (FAS)

3.02

51454

GULP PTB domain containing engulfment adaptor 1 (GULP1)

7.59

3725

Jun proto-oncogene, AP-1 transcription factor subunit (JUN)

3.88

8462

KLF transcription factor 11 (KLF11)

3.24

4170

MCL1 apoptosis regulator, BCL2 family member (MCL1)

2.84

4193

MDM2 proto-oncogene (MDM2)

3.25

10783

NIMA related kinase 6 (NEK6)

2.89

91662

NLR family pyrin domain containing 12 (NLRP12)

3.77

4832

NME/NM23 nucleoside diphosphate kinase 3 (NME3)

3.54

10201

NME/NM23 nucleoside diphosphate kinase 6 (NME6)

2.11

5292

Pim-1 proto-oncogene, serine/threonine kinase (PIM1)

2.62

11040

Pim-2 proto-oncogene, serine/threonine kinase (PIM2)

3.48

415116

Pim-3 proto-oncogene, serine/threonine kinase (PIM3)

3.11

5899

RAS like proto-oncogene B (RALB)

3.72

6461

SH2 domain containing adaptor protein B (SHB)

2.92

30011

SH3 domain containing kinase binding protein 1 (SH3KBP1)

4.36

6881

TATA-box binding protein associated factor 10 (TAF10)

2.10

29844

TCF3 fusion partner (TFPT)

2.20

25816

TNF alpha induced protein 8 (TNFAIP8)

3.09

7188

TNF receptor associated factor 5 (TRAF5)

2.04

8797

TNF receptor superfamily member 10a (TNFRSF10A)

2.15

8795

TNF receptor superfamily member 10b (TNFRSF10B)

2.66

8793

TNF receptor superfamily member 10d (TNFRSF10D)

2.83

51330

TNF receptor superfamily member 12A (TNFRSF12A)

2.28

55504

TNF receptor superfamily member 19 (TNFRSF19)

3.13

3604

TNF receptor superfamily member 9 (TNFRSF9)

3.04

9966

TNF superfamily member 15 (TNFSF15)

7.55

51499

TP53 regulated inhibitor of apoptosis 1 (TRIAP1)

2.11

677

ZFP36 ring finger protein like 1 (ZFP36L1)

3.10

317

apoptotic peptidase activating factor 1 (APAF1)

2.87

8312

axin 1 (AXIN1)

3.27

79444

baculoviral IAP repeat containing 7 (BIRC7)

2.88

23705

cell adhesion molecule 1 (CADM1)

7.51

1116

chitinase 3 like 1 (CHI3L1)

6.56

64651

cysteine and serine rich nuclear protein 1 (CSRNP1)

3.41

26999

cytoplasmic FMR1 interacting protein 2 (CYFIP2)

2.72

1800

dipeptidase 1 (DPEP1)

2.26

1.00E+08

double homeobox 4 (DUX4)

3.16

10913

ectodysplasin A receptor (EDAR)

5.34

79767

engulfment and cell motility 3 (ELMO3)

3.54

2012

epithelial membrane protein 1 (EMP1)

2.31

2014

epithelial membrane protein 3 (EMP3)

3.37

3956

galectin 1 (LGALS1)

3.81

284110

gasdermin A (GSDMA)

4.44

51022

glutaredoxin 2 (GLRX2)

2.93

3002

granzyme B (GZMB)

10.25

2999

granzyme H (GZMH)

2.78

1647

growth arrest and DNA damage inducible alpha (GADD45A)

2.39

4616

growth arrest and DNA damage inducible beta (GADD45B)

2.91

9026

huntingtin interacting protein 1 related (HIP1R)

2.29

3635

inositol polyphosphate-5-phosphatase D (INPP5D)

4.93

3689

integrin subunit beta 2 (ITGB2)

3.10

3429

interferon alpha inducible protein 27 (IFI27)

2.61

2537

interferon alpha inducible protein 6 (IFI6)

2.93

11009

interleukin 24 (IL24)

5.00

79960

jade family PHD finger 1 (JADE1)

3.83

23095

kinesin family member 1B (KIF1B)

3.41

4853

notch receptor 2 (NOTCH2)

2.82

3164

nuclear receptor subfamily 4 group A member 1 (NR4A1)

4.45

5058

p21 (RAC1) activated kinase 1 (PAK1)

2.94

55367

p53-induced death domain protein 1 (PIDD1)

2.83

5551

perforin 1(PRF1)

5.34

5366

phorbol-12-myristate-13-acetate-induced protein 1 (PMAIP1)

2.94

7262

pleckstrin homology like domain family A member 2 (PHLDA2)

2.33

1263

polo like kinase 3 (PLK3)

2.92

5047

progestagen associated endometrial protein (PAEP)

7.01

84306

programmed cell death 2 like (PDCD2L)

2.97

10015

programmed cell death 6 interacting protein (PDCD6IP)

3.55

8682

proliferation and apoptosis adaptor protein 15 (PEA15)

2.27

23645

protein phosphatase 1 regulatory subunit 15A (PPP1R15A)

3.64

5794

protein tyrosine phosphatase receptor type H (PTPRH)

3.69

388

ras homolog family member B (RHOB)

3.08

84236

rhomboid domain containing 1 (RHBDD1)

2.97

55312

riboflavin kinase (RFK)

3.33

6197

ribosomal protein S6 kinase A3 (RPS6KA3)

3.38

117584

ring finger and FYVE like domain containing E3 ubiquitin protein ligase (RFFL)

3.41

54476

ring finger protein 216 (RNF216)

3.08

9262

serine/threonine kinase 17b (STK17B)

2.53

30061

solute carrier family 40 member 1 (SLC40A1)

2.93

10011

steroid receptor RNA activator 1 (SRA1)

2.48

84951

tensin 4 (TNS4)

3.87

7048

transforming growth factor beta receptor 2 (TGFBR2)

3.94

84260

trichoplein keratin filament binding (TCHP)

3.93

8565

tyrosyl-tRNA synthetase 1 (YARS1)

2.86

7429

villin 1 (VIL1)

9.40

64393

zinc finger matrin-type 3 (ZMAT3)

2.86

Revision references:

  1. Heawchaiyaphum, C.; Iizasa, H.; Ekalaksananan, T.; Burassakarn, A.; Kiyono, T.; Kanehiro, Y.; Yoshiyama, H.; Pientong, C. Epstein–Barr virus infection of oral squamous cells. Microorganisms. 2020, 8, 419.
  2. Ahsan, N.; Kanda, T.; Nagashima, K.; Takada, K. Epstein–Barr virus transforming protein LMP1 plays a critical role in virus production. J Virol. 2005;79(7):4415–24.
  3. Chang, Y.; Lee, H. Chang, S.; Hsu, T.; Wang, P.; Chang, Y.; Takada, K.; Tsai, C. Induction of Epstein-Barr Virus Latent Membrane Protein 1 by a Lytic Transactivator Rta. J Virol. 2004, 78, 13028–13036.
  4. Pratt, L.; Zhang, J.; and Sugden, Bill. The latent membrane protein 1 (LMP1) oncogene of Epstein-Barr virus can simultaneously induce and inhibit apoptosis in b cells. J Virol. 2012, 86, 4380–4393.
  5. Heawchaiyaphum, C.; Pientong, C.; Yoshiyama, H.; Iizasa, H.; Panthong, W.; Ekalaksananan, T. General features and novel gene signatures that identify Epstein-Barr virus-associated epithelial cancers. Cancers. 2021, 14, 31.

Reviewer 3 Report

In the manuscript by Heawchaiyaphum, the authors show that established EBV-positive OSCC cells contribute to the progression of OSCC. EBV infection caused mitochondrial stress by reducing the copy number of mitochondrial DNA. The authors showed altered expression of genes associated with glycolysis, including LDHA, GLUT1 and PDK1. This study revealed an oncogenic role for EBV in the progression of OSCC by inducing the Warburg effect and cancer stemness. Results presented by authors indicate that EBV infection induced the expression of CSC marker proteins in a cell type-specific manner by using HSC1 and SCC25 cell lines.

In addition, the xenograft models showed that EBV infection induced OSCC tumorigenesis and promoted tumour growth. The EBV-induced malignant phenotype in OSCC correlated with the upregulation of GLUT1 and LDHA and EBV infection promotes CSC-like properties.

The authors show that EBV infection significantly increased the level of expression of CSC markers: CD44, CD44v6, CD44v9, and CD133 in SCC25-EBV cells, but not in HSC1-EBV cells.

Cancer stem cells in OCSCC express many of the same proteins that are involved in the main ESCs regulatory network. OCT4, NANOG and SOX2 are considered the main regulators of self-renewal and maintenance of the stem cell population in an undifferentiated state.

High levels of expression of OCT4 in OCSCC have been associated with early stage of disease, and better prognosis.

Overexpression of SOX2 has been demonstrated to enhance invasiveness, anchorage-independent growth, and xenotransplantation tumorigenicity in OCSCC cells.

In OSCC and oropharyngeal SCC cell lines, NANOG is overexpressed in the CSC population compared to the parental population.

If the authors turned to the expression of these stemness markers, the study would have convincing evidence of the presence of CSCs and their role in the promotion OSCC stemness.

Methylation of both the host and EBV genomes plays an important role in EBV-associated diseases and should be studied.

Minor remarks

The manuscript contains typos. For example,

Line 121

Minor editing of English language required  

Author Response

  1. If the authors turned to the expression of these stemness markers, the study would have convincing evidence of the presence of CSCs and their role in the promotion OSCC stemness.

Response:  We thank the reviewer for the comments. Our study demonstrated that the infection of EBV induced the plasticity of OSCC by, at least, induction of CD44 and its variant, CD44v6 and CD44v9. Our results are consistent with previous studies that EBV infection induces stemness of EBV-associated epithelial cancers, such as gastric cancer and nasopharyngeal cancer, by up-regulating the CSC marker, CD44, through the expression of EBV oncoproteins, LMP1 and LMP2A [Revision references 1-3].

Previous study reported that CSC markers that were play a role in head-and-neck cancer (HNC) were ALDH1A1, BIM1, CD44, and Lgr5/GPR49 [Revision reference 4]. Therefore, we analyzed the expression of these CSC markers by transcriptomics (Revision Table 1).  As expected, the expression levels of ALDH1A1, CD44, and Lgr5/GPR49 were significantly up-regulated in SCC25-EBV cells. We also found the up-regulation of CSC markers in EBV positive nasopharyngeal carcinoma (NPC: C666-1 and X666) and EBV positive gastric cancer (GC: NUGC3-EBV, SNU484-EBV, SNU484-EBV, SNU638-EBV, SNU638-EBV, SNU719 and YCCEL1) cell lines when EBV-positive cells were compared with EBV-negative cells (Revision Table 1).

In addition, we also analyzed the expression of other CSC markers by transcriptomics. However, the expression levels of POU5F1 (Oct4), NANOG and SOX2 were down-regulated our EBV-infected cell lines. A similar trend of expression was also observed in the EBV-infected NPC and GC cell lines (Revision Table 1).

Revision Table 1. The expression of CSC markers in EBV- associated epithelial cancers.

Gene name

Expression level (Log2(FC))

HSC1-EBV

SCC25-EBV

NPC

GC

ALDH1A1

-1.30

2.18

7.30

-1.00

ANPEP

1.01

2.19

1.02

-0.85

BMI1

3.95

-1.18

2.56

0.81

CD24

-2.36

1.18

1.19

2.37

CD44

-5.66

3.16

-4.34

1.06

FUT4

7.04

-3.41

3.96

2.22

ITGA6

2.80

1.02

-5.48

1.88

KLF4

3.27

-0.89

2.08

1.47

LGR5

-1.71

3.75

8.07

4.96

MSI2

2.71

1.24

0.56

0.84

NANOG

-1.55

ND

0.63

1.20

POU5F1

-2.84

-1.19

-1.01

-0.04

PROM1

-1.80

2.66

8.47

2.60

SALL4

2.73

-0.01

-4.42

-0.91

SOX2

-4.51

-1.30

3.51

-2.10

Remark: ND: Not detect.

Revision references:

  1. Kondo, S.; Wakisaka, N.; Muramatsu, M.; Zen, Y.; Endo, K.; Murono, S.; Sugimoto, H.; Yamaoka, S.; Pagano, J.S.; Yoshizaki, T. Epstein-Barr virus latent membrane protein 1 induces cancer stem/progenitor-like cells in nasopharyngeal epithelial cell lines. J Virol. 2011, 85, 11255-64.
  2. Yang, C.F.; Yang, G.D.; Huang, T.J.; Li, R.; Chu, Q.Q.; Xu, L.; Wang, M.S.; Cai, M.D.; Zhong, L.; Wei, H.J.; Huang, H.B.; Huang, J.L.; Qian, C.N.; Huang, B.J. EB-virus latent membrane protein 1 potentiates the stemness of nasopharyngeal carcinoma via preferential activation of PI3K/AKT pathway by a positive feedback loop. Oncogene. 2016, 35, 3419-31.
  3. Gong, L.P.; Chen, J.N.; Don,g M.; Xiao, Z.D.; Feng, Z.Y.; Pan, Y.H.; Zhang, Y.; Du Y.; Zhang, J.Y.; Bi, Y.H.; Huang, J.T.; Liang, J.; Shao, C.K. Epstein-Barr virus-derived circular RNA LMP2A induces stemness in EBV-associated gastric cancer. EMBO Rep. 2020, 21, e49689.
  4. Zhao, W.; Li, Y.; Zhang, X. Stemness-Related Markers in Cancer. Cancer Transl Med. 2017, 3, 87–95.

  1. Methylation of both the host and EBV genomes plays an important role in EBV-associated diseases and should be studied.

Response:  We thank the reviewer for the comment. In the present study we did not determine the methylation status of both host and EBV genome. However, we further analyzed the expression levels of DNA methyltranferases (DNMTs), including DNMT1, DNMT3B and DNMT3B, that play important roles in the regulation of gene expression. As shown in Revision Table 2, the expression of all DNMTs was up-regulated in HSC1-EBV cell line, while DNMT3A and DNMT3B, but not DNMT1, were up-regulated in SCC25-EBV cell line. Our previous study also demonstrated that DNMT3B induced by EBV infection plays an important role in the progression of OSCC by, at least, inducing the methylation on the miR-145 promoter [Revision reference 1]. Thus, DNMT3B may play important roles in oral carcinogenesis.

Revision Table 2. The expression of CSC markers in EBV- associated epithelial cancers.

Gene name

Expression level (Log2(FC))

HSC1-EBV

SCC25-EBV

NPC

GC

DNMT1

1.28

-1.36

-0.45

0.14

DNMT3A

3.79

1.55

3.31

1.01

DNMT3B

4.12

2.83

0.04

-0.25

Revision reference:

  1. Heawchaiyaphum, C.; Ekalaksananan, T.; Patarapadungkit, N.; Worawichawong. S.; Pientong. C. Epstein-Barr Virus Infection Alone or Jointly with Human Papillomavirus Associates with Down-Regulation of miR-145 in Oral Squamous-Cell Carcinoma. Microorganisms. 2021, 9, 2496.

Minor: The manuscript contains typos. For example, Line 121.

Response:  We thank the reviewer’s kindness. We have corrected the mistake and changed “epression” to “expression” at line 125.
